# GLDS: Global–Local Diversity Selection for Scalable Token Pruning in Vision–Language Models

## Abstract

Transformer-based vision–language models (VLMs) have achieved state-of-the-art performance across a wide range of multimodal tasks, yet their high inference cost remains a major obstacle to scalability. We address the fundamental challenge of efficiently identifying the most informative visual tokens in VLMs—a key bottleneck for large-batch and long-sequence inference. Existing methods often rely on exhaustive or heuristic search strategies that become prohibitively slow or memory-intensive at deployment scale. We introduce **Global-Local Diversity Selection (GLDS)**, a training-free, model-agnostic framework that performs computationally efficient token selection while explicitly balancing local importance with global coverage. To further enhance representational quality under aggressive pruning, GLDS incorporates a determinantal point process (DPP)–based diversity mechanism, ensuring that the retained subset captures both spatially and semantically diverse regions. This leads to consistent improvements across batch sizes and sequence lengths. GLDS accelerates both the prefill and decoding stages, **achieving up to x1.75 speedup in prefill and x1.40 in decoding**, while scaling to inference regimes that overwhelm conventional approaches. On image understanding benchmarks, it maintains performance **with less than 1% absolute accuracy loss**. To our knowledge, this is the first principled and scalable token-selection strategy to achieve a favorable efficiency–accuracy trade-off in VLMs, paving the way for practical deployment of accelerated multimodal transformers.

## 1 Introduction

Large vision-language models (LVLMs) have recently extended the reasoning capabilities of large language models by jointly processing images, videos, and text. These models (e.g. GPT-4, LLaVA-NeXT (Liu et al., 2024a), Qwen2.5-VL (Bai et al., 2025)) typically use a vision encoder (such as CLIP (Radford et al., 2021) or a ViT (Dosovitskiy et al., 2021)) to convert an image into a sequence of visual tokens, which are then concatenated with text tokens and fed into an LLM. However, an image usually yields hundreds to thousands of tokens, far more than a typical text prompt. Since transformer self-attention has quadratic cost in the token sequence length, very long visual sequences dramatically increase computation and memory use. Prior work notes that these dense token sequences "often reach thousands in length, leading to significant computational and memory overhead" (Zhang et al., 2025b). In practice, this makes vision-language inference expensive and slow, especially for high-resolution images or videos. To mitigate this, many methods prune or compress redundant visual tokens during inference. Existing techniques fall into two broad categories: pre-encoder (Zhang et al., 2024a)(reducing tokens during or immediately after the visual encoding) and decoder-stage pruning (Chen et al., 2024b)(dropping or merging tokens during generation). In the pre-encoder stage, one common strategy is attention-based selection (Yang et al., 2024): tokens are scored by their attention weights or relevance and the least-important ones are dropped. Another strategy is feature-similarity pruning (Bolya et al., 2023): visually similar or spatially adjacent tokens are merged or removed to eliminate redundancy. A third approach is text-guided filtering (Zhang et al., 2025e): each visual token's importance is evaluated by its correlation with the language input, e.g. via text-visual attention or mutual information. In the decoder stage, methods may selectively drop tokens or compress the key/value memory during language generation based

on various criteria. These methods can be either training-based or training-free. However, most prior token-pruning work targets architectures like CLIP-based encoders or standard ViTs (Zhang et al., 2024a; Yang et al., 2024; Wang et al., 2024; Liu et al., 2025). State-of-the-art VLMs such as Qwen2.5-VL differ in key ways: they omit a global [CLS] token (commonly used for importance scoring) and include built-in PatchMerger block (Renggli et al., 2022) that already merge 2×2 patches. This limits the applicability of [CLS]-based or uniform downsampling techniques. Moreover, we find that computing full attention or similarity matrices over thousands of tokens can itself be prohibitively expensive. In our empirical analysis on Qwen2.5-VL, we observe that (1) averaging early-layer attention focuses on the first/last patches and yields low variance across tokens, making ranking unreliable, and (2) the overhead of computing token-importance metrics often erases the speedup for single images or large inputs. We also find that aggressive token pruning during decoding gives only marginal latency gains at the cost of large accuracy drops. Motivated by these insights, we propose GLDS, a fast, efficient token-pruning framework that operates only during visual encoding. GLDS progressively removes low-impact visual tokens in the image encoder by exploiting its patch-merging operations, without any extra training. In summary, our contributions are:

1. A comprehensive analysis of existing visual-token pruning methods when applied to Qwen2.5-VL models, highlighting their computational costs and limitations.

2. GLDS, a new training-free pruning framework primarily for Qwen2.5-VL that gradually trims non-informative tokens in the image encoder.

3. Extensive empirical evaluation on multiple image understanding benchmarks, measuring accuracy–throughput trade-offs and how speedup scales with batch size and image resolution.

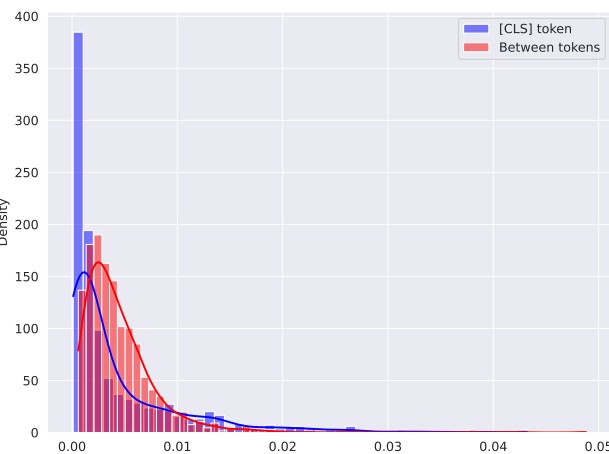

Figure 1: Averaged attention weights density plots.

## 2 RELATED WORKS

### 2.1 TRAINING-AWARE APPROACHES

Several recent methods inject token-compression modules into training. For example, LLaVolta (Chen et al., 2024a), ConvLLaVa (Chunjiang Ge) introduces stage-wise visual-context compression during training or MADTP (Jianjian et al., 2024) uses a learnable dynamic cross-modal alignment. Other works learn special compression schemes: Matryoshka (M3) (Cai et al., 2025), LLaVA-Mini (Zhang et al., 2025d), YOPO (Zhang et al., 2024b).

### 2.2 TRAINING-FREE TOKEN PRUNING (ATTENTION-BASED)

A large family of methods prune tokens at inference time without extra training. Many rely on attention scores to judge token importance: FitPrune (Ye et al., 2024), FastV (Chen et al., 2024b)

or SparseVLM (Zhang et al., 2025e), MustDrop (Liu et al., 2024b) use text-guided attention during the prefilling stage. Similarly, other methods leverage the [CLS] token's attention in the image encoder to drop low-attention patches: FasterVLM (Zhang et al., 2024a), VisionZip (Yang et al., 2024), GlobalCom (Liu et al., 2025), VScan (Zhang et al., 2025a).

### 2.3 TRAINING-FREE TOKEN MERGING (FEATURE-BASED)

Feature-based approaches avoid attention heuristics and merge tokens based on feature similarity: ToMe (Bolya et al., 2023), LLaVA-PruMerge (Shang et al., 2025). Or merges the retaining tokens by semantic similarity to form contextual tokens. The recent AIM (Zhong et al., 2025) performs iterative token merging before the LLM on grouping similar visual embeddings.

### 2.4 MULTI-STAGE METHODS

Several training-free schemes exploit global context or diversity. G-Prune (Jiang et al., 2025) constructs a graph over visual tokens and propagates importance weights; CDPruner (Zhang et al., 2025c) frames token selection as a Determinantal Point Process (DPP) that maximizes conditional diversity given the question, FiCoCo (Han et al., 2025) explicitly splits pruning into three stages: "Filter–Correlate–Compress", PyramidDrop (Xing et al., 2025) segments the LLM layers into stages and drops a fixed fraction of tokens at each stage, iLLaVA (Hu et al., 2024) inserts a fast merging block between layers, VTW (Lin et al., 2025) simply "withdraws" all remaining visual tokens after a chosen deep layer using a KL-divergence criterion. All of these inference-time methods dramatically reduce visual token counts with minimal retraining (if any) by leveraging attention patterns, feature similarity, or token diversity criteria.

## 3 METHODOLOGY

### 3.1 PRELIMINARY: ATTENTION IN VLM ENCODER PHASE

Visual Language Models (VLMs) typically adopt a Transformer-based encoder to process visual tokens. Given an input image, it is partitioned into patches and projected into a sequence of embeddings

$$E_v = \{e_1, e_2, \ldots, e_N\}, \quad e_i \in \mathbb{R}^d, \tag{1}$$

where $N$ is the number of visual tokens and $d$ is the hidden dimension. In practice, $N$ often reaches hundreds or thousands, significantly larger than the number of textual tokens.

Self-attention in the encoder computes pairwise interactions between tokens. For a query matrix $Q \in \mathbb{R}^{N \times d}$, key matrix $K \in \mathbb{R}^{N \times d}$, and value matrix $V \in \mathbb{R}^{N \times d}$, attention weights are defined as:

$$A = \text{Softmax}\left(\frac{QK^\top}{\sqrt{d}}\right), \quad A \in \mathbb{R}^{N \times N}. \tag{2}$$

For high-resolution images in models such as Qwen2.5-VL, $N$ can easily exceed 1,000, making quadratic attention prohibitively expensive in both inference latency and memory consumption.

### 3.2 TOKEN PRUNING PROBLEM FORMALIZATION

Let $E_v = \{e_1, e_2, \ldots, e_N\}$ denote the set of visual tokens produced by the vision encoder, where $N = |E_v|$ is typically much larger than the number of textual tokens. The objective of token pruning is to select a smaller subset $\tilde{E}_v \subseteq E_v$ of size $\tilde{N}$, where $\tilde{N} < N$, while preserving the essential semantic information necessary for accurate prediction.

Formally, token pruning can be viewed as learning a selection mapping function

$$f : E_v \mapsto \tilde{E}_v, \quad \text{with} \quad |\tilde{E}_v| = \tilde{N}. \tag{3}$$

The pruned subset $\tilde{E}_v$ is then passed to the downstream multimodal language model (MLLM). The quality of the pruning strategy can be evaluated by comparing model performance with the original

tokens $E_v$ and with the pruned tokens $\tilde{E}_v$. This can be expressed by a discrepancy measure between their predictive outputs:

$$L\big(\text{model}(E_v),\ \text{model}(\tilde{E}_v)\big), \tag{4}$$

where $L(\cdot, \cdot)$ denotes a suitable loss function (e.g., accuracy drop, cross-entropy divergence, or task-specific metric).

The central challenge lies in designing $f$ such that the selected subset $\tilde{E}_v$ is both *compact* (minimizing computational cost) and *informative* (minimizing task performance degradation).

### 3.3 EMPIRICAL STUDY. CROSS-MODAL TOKEN SELECTION

Empirical analyses show that using text tokens to guide visual token selection is often unreliable. In many multimodal LLMs, the final text token's attention is very dispersed across the image (Xu et al., 2025; Zhang et al., 2025b). In other words, text tokens (like the last instruction token) do not strongly focus on the most relevant visual regions, making it hard to pick important vision tokens from them. As a result, text cues provide little useful signal for which visual tokens to drop.Therefore, many pruning strategies now simply keep all text tokens and only remove visual tokens
To provide further evidence, we simulated the attention distribution of the last text token over visual tokens in Qwen2.5-VL. The distribution is nearly uniform with only few true relevant patches, confirming that the model spreads attention broadly rather than focusing on a group salient patches. Figure 2 shows both the 2D heatmap and projection back to the image grid. The dispersed pattern demonstrates that text tokens do not offer strong localization cues, consistent with the claims in prior work (Xu et al., 2025; Zhang et al., 2025b).
We report an empirical finding that highlights a fundamental limitation in existing methods for ranking attention weights in models advertised as compatible with the Qwen-VL family. A common strategy is to extend the [CLS] token paradigm by averaging attention scores across all tokens and image patches. However, this aggregation dilutes token-level distinctions, leading to a homogenization of ranking weights and ultimately preventing the reliable identification of salient tokens. Figure 1 illustrates the distribution of attention weights under two ranking schemes: (i) alignment with the [CLS] token and (ii) pairwise inter-token ranking. In the [CLS]-based setting, the separation between informative and non-informative tokens is clearly delineated, enabling effective token selection. In contrast, the inter-token ranking distribution collapses into a narrow region, obscuring meaningful boundaries and rendering the method ineffective for relevance estimation.

### 3.4 ATTENTION AND SIMILARITY COMPUTATION COST

A core challenge is that self-attention scales as $O(N^2)$ in the number of tokens. With thousands of visual tokens (e.g. up to 16,384 in Qwen-2.5-VL), the computational and memory cost becomes enormous. In other words, pruning tokens is critical because full attention over a long sequence is prohibitively expensive 3. Some pruning methods compute pairwise similarity between tokens (for diversity-based selection). This requires forming an $N \times N$ cosine-similarity matrix among $N$ tokens. For large $N$, storing and processing this matrix is $O(N^2)$ in memory and time. For example, constructing the cosine similarity matrix $C \in \mathbb{R}^{N \times N}$ and iterating over it can exhaust memory when $N$ is in the thousands Hardware considerations. Modern libraries (like FlashAttention-2 (Dao, 2024)) reduce memory usage of attention to $O(N)$ per layer, but they do not output token-wise importance scores. Thus, many token-pruning methods (which rely on attention scores) must disable these efficient routines to access raw weights. In summary, whether computing attention or pairwise similarities, the operations remain quadratic in $N$, making large-batch or high-resolution inputs very costly.

### 3.5 DECODER-STAGE PRUNING ISSUES

By the time the model is in the LLM (decoder) layers, visual and text information are deeply entangled. Studies indicate that only in the mid-to-late decoder layers do cross-modal interactions become effective (Zhang et al., 2025a). In other words, much of the final prediction relies on visual features that have been propagated through many layers. Pruning tokens at this point is like deleting context that the model has already learned to use, so it disrupts the answer generation. From a complexity standpoint, pruning in the decoder yields only constant-factor gains. The per-token generation step

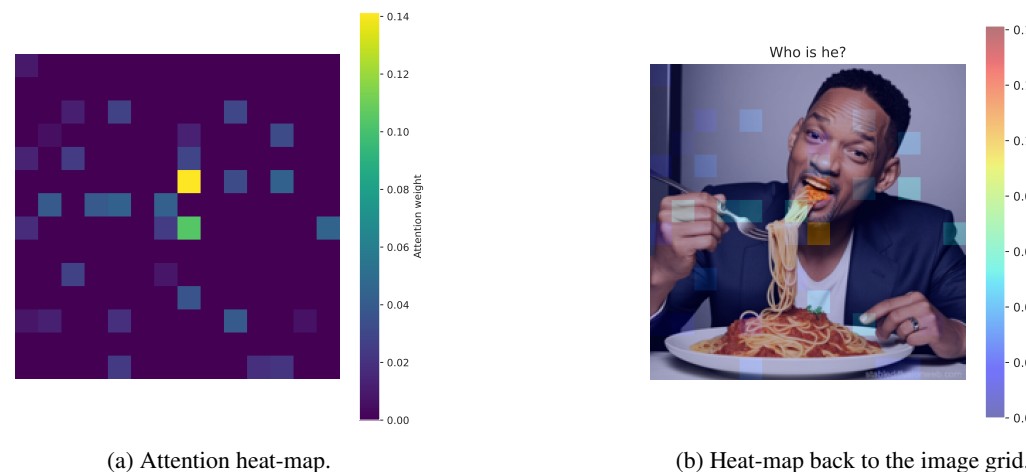

(a) Attention heat-map.

(b) Heat-map back to the image grid.

Figure 2: Attention distribution from last text token to visual tokens (Qwen2.5-VL)

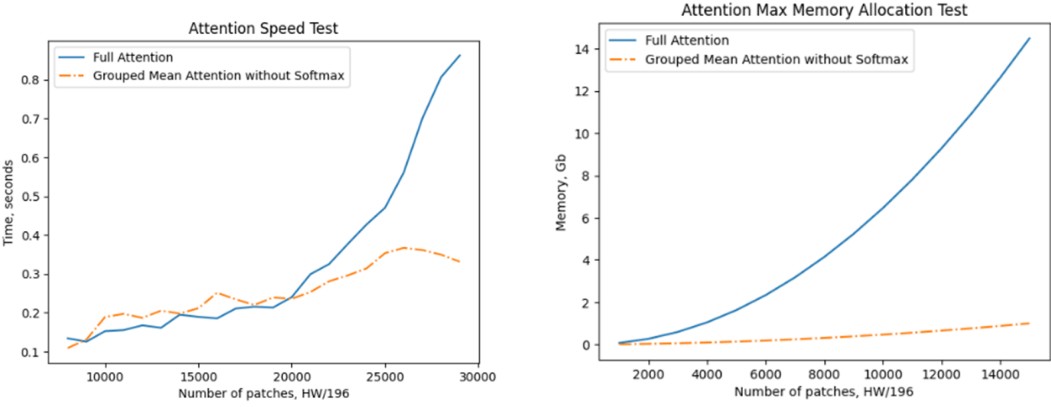

(a) Attention weights calculation time overhead by the number of patches.

(b) Attention weights calculation memory overhead by the number of patches.

Figure 3: Comparison of two attention strategies

still costs $O((N_{\text{vis}} + N_{\text{text}})^2)$ attention. Removing $P$ tokens out of $N$ reduces the constant but does not change the $O(N^2)$ scaling. Moreover, the initial "prefill" step must process all tokens anyway, so any savings accrue only after pruning. In practice, managing the key-value cache when deleting tokens adds overhead. Thus, decoder-stage pruning offers minimal speedup while severely degrading performance.

We did not find existing literature explicitly analyzing pruning at the LLM decoder stage. However, our own experiments show that removing visual tokens during the autoregressive decoding phase dramatically hurts accuracy. In the Table 1 below we provide a empirical results for this observation. Once the optimal set of tokens has been selected at the encoder stage, applying additional pruning at the decoder provides negligible gains in inference speed while causing a notable drop in output quality. This suggests that, for inference efficiency, pruning is most effective when restricted to the encoder, without extending it to the decoder.

## 4 GLOBAL–LOCAL DIVERSE SAMPLING (GLDS)

### 4.1 DPP GLOBAL SCAN (GLOBAL TOKEN SELECTION)

A key component of GLDS is the **global token selection** stage. To solve the problem of ineffectiveness tokens selection based on attention, we aim to retain a diverse yet informative subset of visual

Table 1: Comparison of different pruning strategies on speed and accuracy on TextVQA.

| Model Variant | Speed (tokens/s) | Accuracy (%) |
|---|---|---|
| Base model (Qwen2.5-VL) | 1x | 77.6 |
| Base model + Encoder pruning | 1.15x | 76.7 |
| Base model + Encoder + Decoder pruning | 1.17x | 74.5 |

tokens from the encoder output based on DPP. Given the set of visual tokens $E_v = \{e_1, \ldots, e_M\}$ with $e_i \in \mathbb{R}^d$, our goal is to select a subset $\tilde{E}_v \subseteq E_v$ of size $|\tilde{E}_v| = \tilde{M} \ll M$ that maximizes representational coverage while avoiding redundancy. To this end, we leverage the *determinantal point process* (DPP) as a principled mechanism for subset selection, sharing same assumptions with CDPruner (Zhang et al., 2025c), but on visual tokens only.

**Determinantal Point Processes.** A DPP defines a probability distribution over all subsets $\tilde{E}_v \subseteq E_v$ such that diverse subsets are more likely to be sampled. Formally, for a positive semi-definite kernel matrix $L \in \mathbb{R}^{M \times M}$, the probability of selecting a subset $\tilde{E}_v$ is

$$\mathbb{P}(\tilde{E}_v) \propto \det(L_{\tilde{E}_v}), \tag{5}$$

where $L_{\tilde{E}_v}$ denotes the principal submatrix of $L$ indexed by $\tilde{E}_v$. Intuitively, the determinant measures the volume spanned by the selected vectors, favoring subsets with high diversity.

**Kernel Construction.** In our formulation, the kernel matrix is derived from encoder attention scores. Specifically, we compute a similarity kernel between tokens $e_i$ and $e_j$ as

$$L_{ij} = \alpha \cdot \text{Attn}(e_i, e_j) + (1 - \alpha) \cdot \langle \hat{e}_i, \hat{e}_j \rangle, \tag{6}$$

where $\text{Attn}(e_i, e_j)$ is the normalized attention weight between tokens, $\hat{e}_i = \frac{e_i}{\|e_i\|_2}$ is the $\ell_2$-normalized feature, and $\alpha \in [0, 1]$ balances attention-driven importance and feature-driven similarity.

**Subset Selection.** To obtain a subset $\tilde{E}_v$, we apply $k$-DPP sampling with $k = \tilde{N}$, ensuring that exactly $\tilde{N}$ tokens are selected. In practice, we employ the efficient $k$-DPP algorithm with eigen decomposition of $L$ (Kulesza, 2012), yielding a global set of diverse tokens that balances importance and redundancy.

The selected subset $\tilde{E}_v$ then serves as the input to the subsequent local scan stage, where fine-grained token filtering is applied. See Algorithm 1.

### 4.2 LOCAL TOKENS

To complement the global selection, GLDS conducts a local scan that captures fine-grained details potentially missed by the DPP. We partition the image into a grid of non-overlapping windows (e.g. fixed-size patches) and perform token selection within each window independently at a shallow feature layer. Concretely, select tokens by reshaping attention weights into structured windows and applying fully vectorized top-k operations, enabling simultaneous local selection and global refinement without explicit loops. This ensures that even spatially small but semantically important regions contribute tokens. Formally, if $W_u$ is the set of tokens in window $u$ we choose the highest-scoring tokens in $W_u$ so that the total tokens across all windows meets the desired budget. This localized selection helps prevent missing fine details: as noted in prior works, combining global and local scan mechanisms "selects important tokens based on both local and global information" (Zhang et al., 2025a).

### 4.3 TOKEN MERGING

After selecting the key tokens via global and local scans, GLDS improves token's expressiveness by merging non-retained tokens into retained ones. In essence, any token that is considered redundant

---

**Algorithm 1:** GLDS: Global-Local Diversity Selection

---

**Input:** Aggregated attention weights B $\mathbf{a} \in \mathbb{R}^N$ from attention weights $\mathbb{R}^{N \times N}$, token features
$\quad\quad \mathbf{F} \in \mathbb{R}^{N \times d}$, target number of tokens $k$, local window size $w$
**Output:** Selected token indices $\mathcal{S}$
**Step 1: Pre-filtering (Optional).**
Retain top-$M$ tokens by attention score, $M > k$ (reduces DPP cost).
**Step 2: Compute token quality scores.**
$q_i \leftarrow \sqrt{\frac{\exp(a_i)}{\sum_j \exp(a_j)}} \quad \forall i \in [1, M]$
**Step 3: Compute similarity kernel.**
Normalize token features: $\tilde{f}_i \leftarrow \frac{f_i}{\|f_i\|}$
$K_{ij} \leftarrow \tilde{f}_i^\top \tilde{f}_j \quad \forall i, j \leq M$
**Step 4: Global DPP selection.**
Construct DPP kernel: $L \leftarrow Q^{1/2} K Q^{1/2}$, where $Q = \mathrm{diag}(q_1^2, \ldots, q_M^2)$
Or kernel approximation 6
Run greedy MAP inference (Chen et al., 2018) or faster k-DPP (Gautier et al., 2019) to select
$\quad \mathcal{S}_{\text{global}}$ of size $k_{\text{global}} \leq k$.
**Step 5: Local refinement (window-based top-$k$).**
Partition remaining tokens into non-overlapping windows of size $w$.
**for** *each window $W$* **do**
$\quad\quad$ Select $r$ highest-quality tokens from $W$ according to $q_i$.
$\quad\quad$ Add them to $\mathcal{S}_{\text{local}}$.
**Step 6: Final token set.**
$\mathcal{S} \leftarrow \mathcal{S}_{\text{global}} \cup \mathcal{S}_{\text{local}}$
**return** $\mathcal{S}$

---

compared to the remaining token is merged with the closest important token in terms of distance, and its contribution to the embedding is added.

Formally, let $E_v \in \mathbb{R}^{M \times d}$ denote the set of visual token embeddings after global and local pruning, and let $\tilde{E}_v \subseteq E_v$ be the retained subset of size $T = |\tilde{E}_v|$. We define a binary mask $\mathbf{m} \in \{0, 1\}^M$ indicating retained tokens:

$$m_i = \begin{cases} 1, & e_i \in \tilde{E}_v \\ 0, & \text{otherwise} \end{cases}. \tag{7}$$

Let $R = \tilde{E}_v$ be the retained token embeddings and $N = E_v \setminus R$ the non-retained embeddings. We compute pairwise cosine similarities between non-retained and retained tokens:

$$S = \text{cosine\_sim}(N, R) \in \mathbb{R}^{(M-T) \times T}, \tag{8}$$

and assign each non-retained token $n_i \in N$ to its nearest retained token $r_j \in R$ via

$$j^\star = \arg\max_j S_{ij}. \tag{9}$$

The merged feature embeddings are then computed by aggregating each retained token with all assigned non-retained tokens:

$$\tilde{r}_j = \frac{r_j \cdot \text{scaling} + \sum_{i:j^\star = j} n_i}{\text{scaling} + |\{i : j^\star = j\}|}, \tag{10}$$

where scaling $\geq 1$ balances the contribution of the original retained token.

This approach effectively reduces redundancy while preserving semantic content, and it is fully differentiable and training-free. The resulting set of merged embeddings $\tilde{R} = \{\tilde{r}_1, \ldots, \tilde{r}_T\}$ is then passed to the decoder or subsequent stages of the VLM pipeline.

## 4.4 ENGINEERING IMPROVEMENTS

**Attention weights computation.** Due to the architectural characteristics of image processing in batch inference and the computational requirements of attention weight-based pruning, modern in-

ference systems utilizing FlashAttention-2 and XFormers (Lefaudeux et al., 2022) incur substantial overhead during the computation of attention matrices (Zhang et al., 2025b;c; Yang et al., 2024). The problem becomes particularly acute in models without a [CLS] token, where the absence of a dedicated global representation requires computing the entire attention weight matrix, resulting in substantially higher computational cost compared to evaluating only a single row. To solve this problem we implement method for computing attention weights, comprising pre-aggregating query and key embeddings into spatial blocks prior to attention, thereby reducing attention computation from quadratic time and memory $O(N^2)$ to $O((N/k)^2)$, where $k$ is the block size, thus achieving block-level efficiency while preserving averaged attention signals. Proposition 1 shows that the grouped approximation is close to the true mean attention under local smoothness assumptions.

**Efficient Token Merging.** To find nearest tokens, a cosine similarity is usually used, which is effectively computing all pairwise similarities in a fairly memory-heavy way. When PyTorch computes cosine similarity, it broadcasts the shapes, that means a huge temporary tensor of size $(N - T) \times T \times D$ is created in memory. So instead of recomputing norms for every pair we just compute them once for each token and then use a dot product for all pairs. We use efficient CUDA kernels and nearest-neighbor heuristics to quickly assign merge candidates. Inspired by QuickMerge++ (Liu & Yu, 2025), we can weight merges by token "saliency" or norm so that less-important tokens donate their mass to stronger ones. The merging is done via batch matrix operations, making it very fast. We also allow a dynamic merge ratio: larger images or simpler scenes can merge more tokens.

## 5 EXPERIMENTS

### 5.1 EXPERIMENTAL SETUP

All experiments are conducted using the Qwen-2.5-VL model as the primary backbone, given its state-of-the-art performance in image understanding tasks and unique architectural challenges for token pruning. Inference is executed on NVIDIA A100 GPUs with 80GB of memory, using FlashAttention-2 as the default attention implementation.

We emphasize that, to the best of our knowledge, there are currently no widely available and provable open-source implementations of token pruning that support the Qwen-2.5-VL architecture effectively. At the time of writing, only two works explicitly target Qwen2.5-VL models: **VScan**, which provides an open-source codebase, and **BTP (Balanced Token Pruning)** Li et al. (2025), which has no released implementation. Consequently, the majority of our comparisons are conducted against VScan. Beyond Qwen-2.5-VL, we also evaluate GLDS on LLaVA and LLaVA-Next models. Although the core implementation of **GLDS** is designed to address the lack of a [CLS] token and the consequent limitations of attention-based averaging in Qwen2.5-VL, these experiments demonstrate that the algorithm also generalizes effectively to LLaVA-like architectures.

In our tests all pruning methods were set on the 75% pruning ration for the LLaVa-like models and 60% for the Qwen2.5-VL. Other studies and examples are located in Appendix D.

### 5.2 BENCHMARKS

Quality evaluations are performed on standard multimodal benchmarks including MME (Fu et al., 2023), GQA (Hudson & Manning, 2019), POPE (Yifan Li & Wen, 2023), and TextVQA (Singh et al., 2019).

### 5.3 RESULTS AND ANALYSIS

**Inference Quality.** The results in Table 2 indicate that, within the Qwen2.5-VL model class, the GLDS method achieves the highest quality at the specified token pruning level. Comparable performance is also observed for the Llava model class, demonstrating the method's effectiveness across different architectures.

**Inference Efficiency.** Finally in Table 3, we provide an ablation study analyzing the theoretical prefill and decoding speedups of GLDS under varying batch sizes, in addition to its empirical per-

Table 2: Comparison of pruning methods across different models (Qwen2.5-VL, LLaVA, LLaVA-Next) on standard benchmarks. Benchmarks include MME, GQA, POPE, and TextVQA. Bold indicates the best method per model.

| Model | Method | MME | GQA | POPE | TextVQA |
|-------|--------|-----|-----|------|---------|
| Qwen2.5-VL | Baseline | 2325 | 61.9 | 86.6 | 77.6 |
|  | VScan | – | 60.9 | 85.9 | 76.1 |
|  | **GLDS (Ours)** | – | **61.1** | **86.1** | **76.7** |
| LLaVA | Baseline | 1861 | 61.9 | 85.9 | 58.2 |
|  | VTC-CLS | 1735 | 58.8 | **87.1** | 57.0 |
|  | VisionZip | 1761 | 57.6 | 83.2 | 55.5 |
|  | Matryoshka (trained) | 1731 | **62.5** | 87.0 | 56.8 |
|  | VScan | 1781 | 59.1 | 84.2 | 56.1 |
|  | **GLDS (Ours)** | **1821** | 59.6 | 84.3 | **57.2** |
| LLaVA-Next | Baseline | 1844 | 61.3 | 86.5 | 61.3 |
|  | VisionZip | **1845** | 62.5 | **87.9** | 60.2 |
|  | Matryoshka (trained) | 1821 | **63.6** | 87.7 | 60.9 |
|  | VScan | 1842 | 62.7 | 87.2 | 60.8 |
|  | **GLDS (Ours)** | 1844 | 63.0 | 87.5 | **61.1** |

Table 3: Comparison of inference time (in milliseconds per step) between **Qwen-VL-2.5** and **GLDS** across batch sizes. We report both *prefill* (first forward pass) and *decoding* (autoregressive generation) times. Lower is better.

| Batch Size | Prefill Time (ms) | | Decoding Time (ms) | |
|------------|---------|------|---------|------|
|  | Qwen-VL | GLDS | Qwen-VL | GLDS |
| 1 | 82.3 | 52 (1.58x) | 30 | 29 (1.03x) |
| 2 | 160 | 100 (1.6x) | 40 | 39 (1.03x) |
| 4 | 350 | 200 (1.75x) | 41 | 40 (1.03x) |
| 8 | 700 | 400 (**1.75x**) | 51 | 43 (1.18x) |
| 16 | 1400 | 1000 (1.45x) | 80 | 60 (1.33x) |
| 32 | 2900 | 2000 (1.45x) | 140 | 100 (1.4x) |
| 64 | 5800 | 4000 (1.45x) | 260 | 185 (**1.41x**) |

formance. This allows us to quantify both the measured efficiency improvements and the potential upper bounds of acceleration in large-sequence or large-batch scenarios.

## 6 CONCLUSION

In this work, we introduced **Global-Local Diversity Selection (GLDS)**, a training-free and model-agnostic framework for efficient visual token reduction in vision–language models. GLDS leverages a determinantal point process (DPP) to perform a principled global scan, ensuring diversity and coverage among selected tokens, while complementing it with a lightweight local scan to capture fine-grained visual details. Furthermore, GLDS integrates a memory-efficient token merging mechanism and several engineering optimizations, enabling scalability across larger batch sizes and high-resolution inputs. Empirical results demonstrate that GLDS consistently accelerates inference while preserving near-lossless accuracy across multiple benchmarks, **with speedups reaching up to x1.75 in the prefill stage and x1.4 in decoding for larger batch settings.** Importantly, GLDS extends pruning support to the Qwen-VL family, where previous methods were either inapplicable or inefficient. Overall, GLDS provides a practical, scalable, and effective solution for accelerating vision–language inference, and establishes a foundation for future exploration of principled diversity-based token selection methods in multimodal learning.

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

## A  USAGE OF LLMS

During the preparation of this work, large language models (LLMs) were employed solely to assist with language polishing and grammar correction. No parts of the research design, experimental setup, analysis, or interpretation relied on LLMs. All technical contributions, algorithmic designs, and empirical evaluations presented in this work are original and conducted entirely by the authors.

# B  AGGREGATED ATTENTION WEIGHTS.

In VLM token pruning, it is common to aggregate attention weights to estimate token importance. When a model includes a `[CLS]` token, the aggregated attention for token $i$ is typically computed as the mean of attention weights from all heads directed toward the `[CLS]` token:

$$\bar{a}_i = \frac{1}{H} \sum_{h=1}^{H} \mathrm{Attn}_h(i, \texttt{[CLS]}),$$

where $H$ is the number of attention heads. In models without a `[CLS]` token, the aggregation is performed across all token pairs by averaging the attention weights over heads and then summing over the query tokens:

$$\bar{a}_i = \frac{1}{H} \sum_{h=1}^{H} \sum_{j=1}^{N} \mathrm{Attn}_h(i, j),$$

where $N$ is the total number of visual tokens. This aggregation provides a scalar importance score per token, used in subsequent global or local selection stages.

# C  GROUPED ATTENTION VIA TOKEN PRE-AGGREGATION

## C.1  GROUPED QUERY-KEY REPRESENTATION

We consider an input sequence of $S$ tokens, with multi-head self-attention of $H$ heads and head dimension $d$. Let $Q, K, V \in \mathbb{R}^{S \times d}$ be the query, key, and value matrices for one head (we omit the head index for clarity). In standard scaled dot-product attention, we form the score matrix $X = QK^\top \in \mathbb{R}^{S \times S}$ (scaled by $1/\sqrt{d}$) and compute the attention weights via row-wise softmax:

$$A_{i,j} = \frac{\exp\left(Q_i^\top K_j / \sqrt{d}\right)}{\sum_{m=1}^{S} \exp\left(Q_i^\top K_m / \sqrt{d}\right)}. \tag{11}$$

Thus a single head requires $O(S^2 d)$ operations to compute $QK^\top$ and $O(S^2)$ space to store the weight matrix. In particular, with $H$ heads the cost is $O(HS^2 d)$, and storing all attention matrices costs $O(HS^2)$ memory, which becomes prohibitive for large $S$.

## C.2  GROUPED QUERY-KEY REPRESENTATION

To reduce this cost, we partition the sequence into $M$ groups of size $G$ (assume $G$ divides $S$, so $M = S/G$). For group $g = 1, \ldots, M$, let indices $(g-1)G + 1, \ldots, gG$ belong to group $g$. We define the *group-averaged* queries and keys for each group as

$$\bar{Q}_g = \frac{1}{G} \sum_{i=(g-1)G+1}^{gG} Q_i, \qquad \bar{K}_g = \frac{1}{G} \sum_{i=(g-1)G+1}^{gG} K_i,$$

so that $\bar{Q}, \bar{K} \in \mathbb{R}^{M \times d}$ collect these means. We then perform attention among these $M$ group-representatives:

$$\widetilde{A} = \mathrm{softmax}\left(\bar{Q}\,\bar{K}^\top / \sqrt{d}\right) \in \mathbb{R}^{M \times M}. \tag{12}$$

Each query token in group $g$ can then use $\widetilde{A}_{g,h}$ as the attention weight to tokens in group $h$. Intuitively, if tokens within a group have similar keys and queries (local smoothness), then averaging is a reasonable surrogate. This "early grouping" reduces the attention problem from $S$ tokens to $M = S/G$ group-representatives.

## C.3  APPROXIMATION GUARANTEE

Let $A_{i,j}$ be the full attention weight from token $i$ to $j$ in the standard model, and define the *group-averaged* true weight from group $g$ to group $h$ as

$$\bar{\alpha}_{g,h} = \frac{1}{G} \sum_{i \in g} \sum_{j \in h} A_{i,j},$$

i.e. the average attention mass that queries in group $g$ place on keys in group $h$. Our goal is to show $\widetilde{A}_{g,h} \approx \bar{\alpha}_{g,h}$ under mild conditions.

**Proposition 1.** *Suppose that within each group $g$, all $Q_i$ (resp. $K_j$) are close to their mean $\bar{Q}_g$ (resp. $\bar{K}_h$), so that*

$$\max_{i \in g} \|Q_i - \bar{Q}_g\| \leq \epsilon_Q, \qquad \max_{j \in h} \|K_j - \bar{K}_h\| \leq \epsilon_K.$$

*Then the group-level attention weight satisfies*

$$\left| \widetilde{A}_{g,h} - \bar{\alpha}_{g,h} \right| \leq C\left(\epsilon_Q + \epsilon_K\right),$$

*for some absolute constant $C > 0$, i.e. the error is bounded by the within-group variances scaled by the Lipschitz constant of softmax.*

*Proof.* By linearity of sums, the group-mean dot-product can be written as

$$\bar{Q}_g^\top \bar{K}_h = \frac{1}{G^2} \sum_{i \in g} \sum_{j \in h} Q_i^\top K_j.$$

Hence the score used in the grouped attention $\widetilde{A}_{g,h}$ equals the average of the true pairwise scores (up to the $1/\sqrt{d}$ scaling).

Now, for each $i \in g$ and $j \in h$, we have

$$|Q_i^\top K_j - \bar{Q}_g^\top \bar{K}_h| \leq \|Q_i - \bar{Q}_g\| \cdot \|K_j\| + \|\bar{Q}_g\| \cdot \|K_j - \bar{K}_h\| \leq O(\epsilon_Q + \epsilon_K).$$

Thus each individual score deviates from the group mean score by at most $O(\epsilon_Q + \epsilon_K)$.

Since the softmax function $\sigma : \mathbb{R}^M \to \mathbb{R}^M$ has Jacobian

$$J_\sigma(z) = \operatorname{diag}(\sigma(z)) - \sigma(z)\sigma(z)^\top,$$

its spectral norm satisfies $\|J_\sigma(z)\|_2 \leq 1/2$. Therefore $\operatorname{softmax}(\cdot)$ is $1/2$-Lipschitz in $\ell_2$, i.e.

$$\|\operatorname{softmax}(x) - \operatorname{softmax}(y)\|_2 \leq \tfrac{1}{2}\|x - y\|_2.$$

Applying this to the score vectors of group-averaged vs. true tokens yields

$$|\widetilde{A}_{g,h} - \bar{\alpha}_{g,h}| \leq C(\epsilon_Q + \epsilon_K).$$

Finally, since averaging over heads and tokens is linear, this error bound carries over directly to the final $N/G$-dimensional score vector obtained after aggregation. $\qquad\square$

# D ANALYSIS AND ADDITIONAL EXPERIMENTS

To complement our large-scale benchmark evaluation, we provide additional studies that illustrate the practical behavior of GLDS under diverse conditions.

**Time Overhead Breakdown.** We first decompose the end-to-end inference time into individual components, including attention weight computation, global DPP-based selection, local window refinement, and token merging. Results are reported across batch sizes 1, 4, 16, and 32 (Table 4). The dominant cost remains the local scan and token merging computations, while the global scan overhead is negligible due to our use of a top-$k$ warm-up approximation. Attention contributes minimal runtime, showing that GLDS overhead does not bottleneck scaling.

**Retention Ratio Sensitivity.** We next study the effect of varying token retention ratios on TextVQA under two temperature settings ($T = 0$ and $T = 1$). Results in Table 5 show that GLDS degrades gracefully, with less than $1\%$ accuracy loss even at $40\%$ retention. Only at extreme compression ($10\%$) do we observe a notable drop, demonstrating robustness of GLDS to aggressive pruning.

Table 4: Overhead breakdown of GLDS (ms). Global scan overhead is negligible due to top-$k$ warm-up.

| Operation | Batch 1 | Batch 4 | Batch 16 | Batch 32 |
|---|---|---|---|---|
| Attention | 0.35 | 1.6 | 7 | 14 |
| Local Scan | 0.8 | 3.3 | 13 | 26 |
| Token Merging | 0.5 | 2.3 | 9 | 18 |
| Total GLDS Overhead | 1.7 | 7.2 | 29 | 58 |

Table 5: Accuracy (%) of GLDS vs. Qwen2.5-VL baseline on TextVQA across different retention ratios.

| Method | 75% | 60% | 50% | 40% | 33% | 25% | 10% |
|---|---|---|---|---|---|---|---|
| Baseline (Qwen2.5-VL, T=0) | 77.6 | 77.6 | 77.6 | 77.6 | 77.6 | 77.6 | 77.6 |
| GLDS (T=0) | 77.5 | 77.4 | 77.1 | 76.6 | 76.1 | 74.6 | 65.9 |
| Baseline (Qwen2.5-VL, T=1) | 72.4 | 72.4 | 72.4 | 72.4 | 72.4 | 72.4 | 72.4 |
| GLDS (T=1) | 72.0 | 72.5 | 71.7 | 71.5 | 70.5 | 68.9 | 59.4 |

**High-Resolution Images.** Benchmarks such as TextVQA often contain relatively small input images, underestimating the potential benefits of pruning. To reflect deployment scenarios, we evaluate GLDS on large-resolution images ($2940 \times 1960$ pixels). Results in Table 6 demonstrate that GLDS achieves substantial e2e acceleration at batch sizes 8, 16, and 32 with Aggressive pruning strategy (98%), with more than $4\times$ speedup at scale, while maintaining nearly identical accuracy in answers D.1.

## D.1 VISUALIZATION

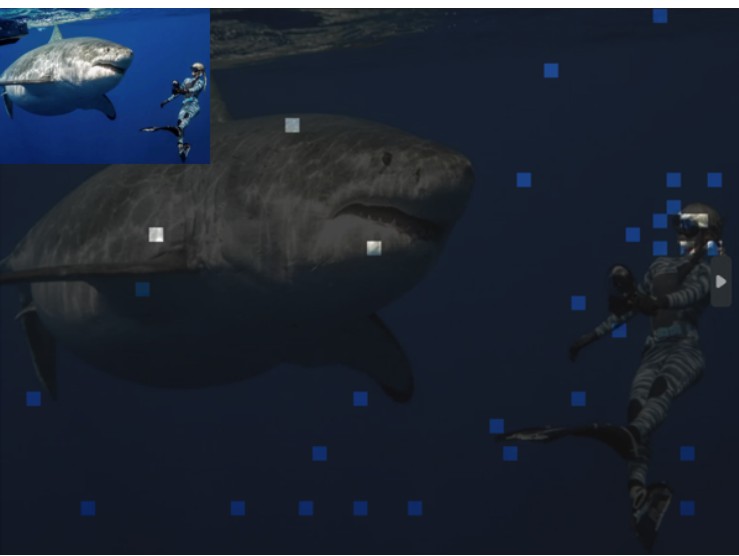

Figure 4:
Why is this so funny?
Original: person in a diving suit, which is typically used for swimming
Pruned: person in a zebra-striped wetsuit

Table 6: Acceleration on large-resolution images up to (3000 × 2000).

| Model | Batch 8 | Batch 16 | Batch 32 |
|---|---|---|---|
| Qwen2.5-VL (Baseline) | 16.2 s | 30.6 s | OOM |
| GLDS (2%) | 4.3 s | 7.1 s | 10.5 s |
| Speedup (2%) | 3.76× | 4.31× | – |

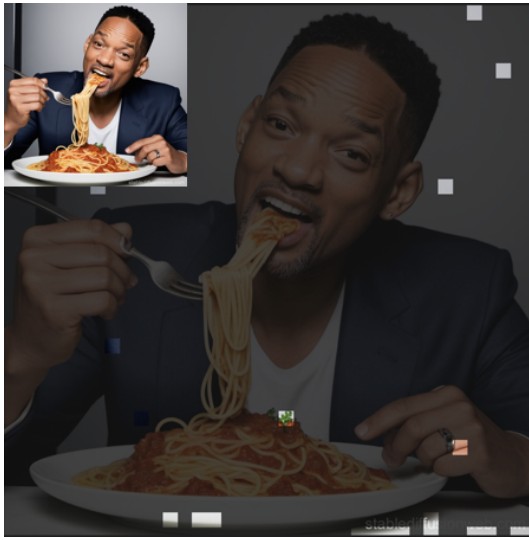

Figure 5:
Who is he?
Original: Will Smith
Pruned: person eating spaghetti with a fork. The text at the bottom of the image reads "stablediffusionweb.com," which suggests that this image might have been generated

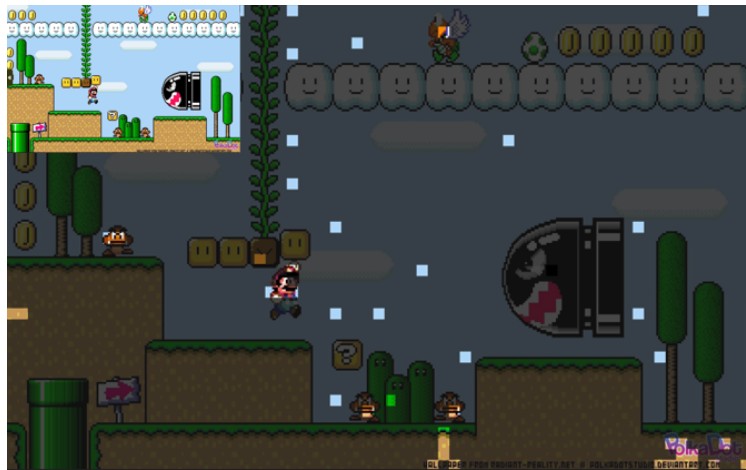

Figure 6:
Who is his best friend?
Original: his best friend is Luigi
Pruned: his best friend is Luigi

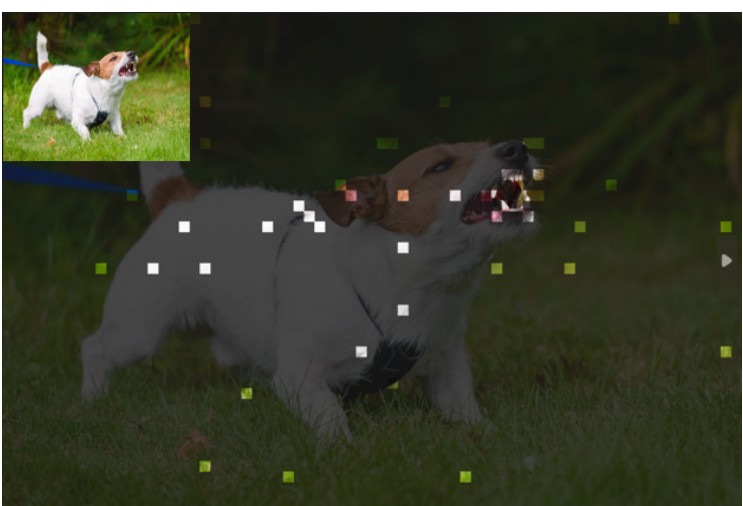

Figure 7:
Is this an AI generated one?
Original: it's not possible to definitively determine if the image is artificially generated
Pruned: The image you provided appears to be an artificial creation, likely generated by AI

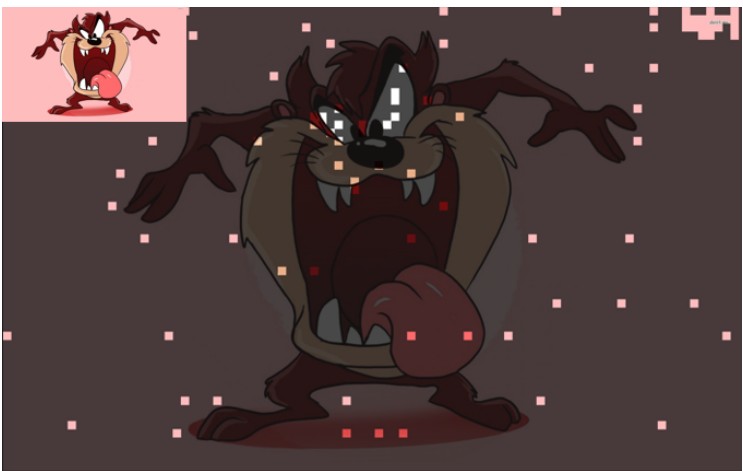

Figure 8:
Who is his best friend?
Original: Bugs Bunny
Pruned: Bugs Bunny

