# OpenReview forum: "GLDS: Global–Local Diversity Selection for Scalable Token Pruning in Vision–Language Models"
_ICLR.cc/2026/Conference — ICLR 2026 Conference Withdrawn Submission_

### Official Review · Reviewer_CKsz · 2025-10-27

**Soundness:** 2
**Presentation:** 1
**Contribution:** 1
**Rating:** 2
**Confidence:** 4

**Summary:**

The authors propose a token pruning technique called GLDS that combines determinantal point process-based global selection with local window refinement to reduce visual tokens in vision-language models during inference. The method is specifically designed to address limitations in modern architectures like Qwen2.5-VL that lack [CLS] tokens, achieving up to 1.75x speedup while maintaining accuracy within 1% on benchmarks.

**Strengths:**

-  The paper shows speedups with minimum accuracy loss.
- The method is training-free and model-agnostic.
-  The authors appear to be releasing code and provide enough implementation detail that someone could actually reproduce this.

**Weaknesses:**

- Incorrect quotation marks throughout, low-resolution compressed figures.
- Table 2 bolding error: baseline (61.9) outperforms GLDS (61.1) but GLDS is bolded as "best".
- Undefined parameters: "scaling" in Eq. 10 is never explained.
- No comparison with ToMe, AIM, PruMerge despite citing them.
- Method is just DPP (from CDPruner) + local windows (from VScan) + merging (from ToMe).
- Claims of "first principled approach" are false given extensive prior work.
- Sections 3.1-3.3 waste space explaining basic transformer attention everyone knows.
- Different retention ratios across models (60% vs 75%) may suggest cherry-picking.
- No error bars or statistical significance testing.
- No proper ablation isolating each component's contribution.

**Questions:**

- What happens if you replace DPP with simpler alternatives (random sampling, k-means clustering, or furthest point sampling)? This ablation would clarify if the complexity of DPP is justified.
- Why do you use different retention ratios (60% for Qwen2.5-VL vs 75% for LLaVA)? Please provide results with uniform retention ratios across all models.
- Can you provide comparisons with ToMe, AIM and PruMerge? If implementation challenges exist, please explain specifically what prevents fair comparison.
- Table 2 shows GLDS (61.1) underperforming baseline (61.9) on GQA for Qwen2.5-VL, yet GLDS is bolded. Is this an error or is there a different interpretation?
- What is the "scaling" parameter in Equation 10? How is it set and what is its sensitivity?
- What is the end-to-end speedup for complete inference pipelines including preprocessing/postprocessing?
- All results appear to be single runs. Can you provide error bars or confidence intervals across multiple seeds?
- For the ablation study, what is the individual contribution of each component (DPP alone, local windows alone, merging alone)?

---

### Official Review · Reviewer_nB9D · 2025-10-31

**Soundness:** 2
**Presentation:** 2
**Contribution:** 2
**Rating:** 2
**Confidence:** 4

**Summary:**

This paper introduces Global-Local Diversity Selection (GLDS), a training-free visual token pruning framework that balances local importance and global coverage to accelerate VLM inference. Experimental results demonstrate that GLDS achieves strong performance retention while delivering significantly higher efficiency, outperforming existing state-of-the-art methods.

**Strengths:**

1. The proposed approach achieves advantegeous performance-efficiency trade-off.
2. The structure of this paper is clear.

**Weaknesses:**

1. The novelty of this paper is limited. The proposed complementary global-local selection strategy for visual tokens closely resembles that of VScan, and the underlying motivation appears largely similar.
2. The experimental results are not comprehensive. The performance is evaluated on only four standard benchmarks with a single predefined pruning ratio. Additional experiments under diverse settings are needed to better validate the effectiveness of the proposed approach.
3. The motivation of this work should be better clarified. Most of the empirical analyses presented have already been explored in previous studies.
4. The presentation of this paper is weak, and the overall work reads more like a technical report than a research paper. Most of the contributions appear to be engineering-oriented, offering limited conceptual or theoretical insights to advance the field.

**Questions:**

See the weaknesses section above.

---

### Official Review · Reviewer_64Sj · 2025-11-01

**Soundness:** 2
**Presentation:** 3
**Contribution:** 2
**Rating:** 4
**Confidence:** 4

**Summary:**

This paper proposed a training-free and model-agnostic token pruning framework for vision-language models, which targets to accelerate the VLMs with PatchMerger and without [cls] like Qwen2.5-VL. The method is composed of a DPP-based diversity mechanism with local Top-K selection and token merging of the dropped tokens. Extensive experiments on different benchmarks demonstrate the effectiveness of the proposed method.

**Strengths:**

1. The analysis of the limitations for deploying existing methods to Qwen2.5-VL sounds good and insightful.
2. The proposed method can achieve speedup on both the prefill and decode stages, which is beneficial to the deployment.
3. The paper is well-written and easy to follow.

**Weaknesses:**

1. A comprehensive ablation study is required, e.g., $\alpha$, Top-M, w, r, etc.
2. Compared with previous DPP-based methods, the main contribution of this work is adapting DPP to the models without [CLS] like Qwen2.5-VL, which I think is incremental in terms of novelty.
3. The experimental comparisons are insufficient, as the authors mainly compared with VScan. However, works like FitPrune/FastV/SparseVLM/VTW are not compared, even on LLaVA.
4. Some important references need to be compared or discussed, e.g., Dynamic-LLaVA, CoreMatching.
5. Some claims are not self-contained, e.g., although the authors doubt the reliability of the attention score in Qwen2.5-VL, they still introduce the attention score into DPP kernel design.

[1] Dynamic-LLaVA: Efficient Multimodal Large Language Models via Dynamic Vision-language Context Sparsification. ICLR 2025.
[2] CoreMatching: A Co-adaptive Sparse Inference Framework with Token and Neuron Pruning for Comprehensive Acceleration of Vision-Language Models. ICML 2025.

**Questions:**

See weaknesses.

---

### Note · Authors · 2025-11-25

I have read and agree with the venue's withdrawal policy on behalf of myself and my co-authors.